# Enhancing Drones for Law Enforcement and Capacity Monitoring at Open Large Events

**Pablo Royo** †, **Àlex Asenjo** †, **Juan Trujillo** †, **Ender Çetin** † and **Cristina Barrado** *,†

Computer Architecture Department, Technical University of Catalonia, UPC BarcelonaTech, 08860 Castelldefels, Spain

* Correspondence: cristina.barrado@upc.edu
† These authors contributed equally to this work.

**Abstract:** Police tasks related with law enforcement and citizen protection have gained a very useful asset in drones. Crowded demonstrations, large sporting events, or summer festivals are typical situations when aerial surveillance is necessary. The eyes in the sky are moving from the use of manned helicopters to drones due to costs, environmental impact, and discretion, resulting in local, regional, and national police forces possessing specific units equipped with drones. In this paper, we describe an artificial intelligence solution developed for the Castelldefels local police (Barcelona, Spain) to enhance the capabilities of drones used for the surveillance of large events. In particular, we propose a novel methodology for the efficient integration of deep learning algorithms in drone avionics. This integration improves the capabilities of the drone for tasks related with capacity control. These tasks have been very relevant during the pandemic and beyond. Controlling the number of persons in an open area is crucial when the expected crowd might exceed the capacity of the area and put humans in danger. The new methodology proposes an efficient and accurate execution of deep learning algorithms, which are usually highly demanding for computation resources. Results show that the state-of-the-art artificial intelligence models are too slow when utilised in the drone standard equipment. These models lose accuracy when images are taken at altitudes above 30 m. With our new methodology, these two drawbacks can be overcome and results with good accuracy (96% correct segmentation and between 20% and 35% mean average proportional error) can be obtained in less than 20 s.

**Keywords:** crowd counting; deep learning; object detection; prediction; security

## 1. Introduction

The main objective of law enforcement agents is the protection of citizens, public infrastructure, and governmental institutions. Drones have been shown to be a very useful asset for all these tasks and are very cost effective. For these reasons, police eyes-in-the-sky are transitioning from manned helicopters to drones. Drones also reduce the environmental impact and increase the discretion of surveillance.

During the COVID-19 pandemic, many local, regional, and national police forces started experimenting with drones. Drones limit physical contact between humans and avoid the spread of the virus. Additionally, during the curfew streets were nearly empty and ground risks could be better mitigated. In addition, many countries regulation have been put in place to support emergency responders and police crews in operating drones more easily.

Since 2020, many law enforcement agencies have created specific units equipped with drones. In some cases, drone operations are subcontracted to private security operators. In both situations, a small fleet of drones is supporting aerial surveillance at political demonstrations, large sporting events, or summer festivals.

In this paper, we describe the deep learning software solution developed for the Castelldefels local police (Barcelona, Spain) to enhance the capabilities of drones used for

surveillance of large events. We developed an on-site solution for them to count the number of persons seen from the drone. Results show that the current artificial intelligence (AI) models reach their limits for small devices and at altitudes above 30 m. At high altitudes, detection using state-of-the-art models results in low accuracy and, thus, new methods are needed.

Up to 18 datasets with images and videos for crowd counting are well documented in [1] with all data open licensed under Creative Commons. For instance, the popular Shanghai dataset has hundreds of labelled images [2]. It has two parts, A and B, one with highly congested images and the other with sparse people. Nevertheless, most images in these datasets are taken at ground level or at the altitude of a surveillance camera. Both are lower than typical drone altitude.

Wen et al. created the Drone Crowd dataset and published it together with their detection algorithm [3]. Although the Drone Crowd dataset has images taken from an altitude similar to the altitude used by law enforcement, the angle of vision is zenithal. In our experience with police tasks, they usually apply some slanted angle to their drone cameras. There are different reasons for this: in general to avoid the safety risks of hovering above a crowd, in other cases to avoid being seen, and in most cases, to obtain a global view of the whole area. Oblique images are better to quickly give an overview of the situation because they capture the context and any reference point of the geographical area. A motivating example of the type of images we aim to process is shown in Figure 1. One can observe the highly different densities of people in the image. This type of image is also known as uneven distribution.

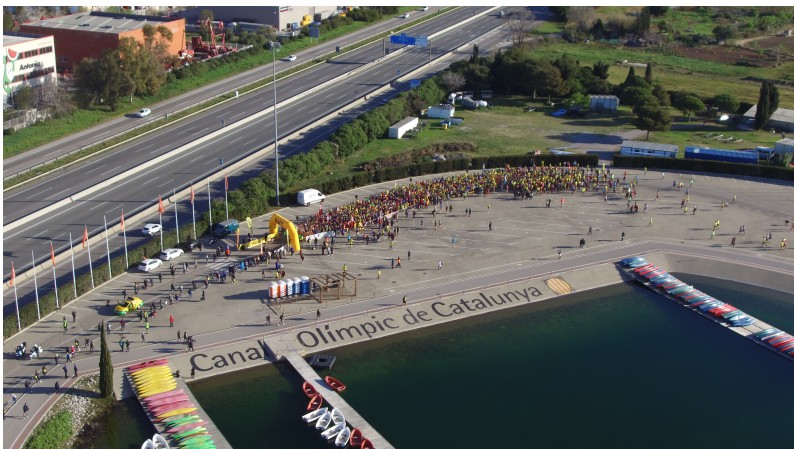

**Figure 1.** Image of the start of the Mediterranean Marathon at the Canal Olímpic de Catalunya, Castelldefels, Spain, 20 March 2020 (source: Policia Local de Castelldefels, permission granted).

The research question that we aim to answer is: "Is it possible to develop a solution to count persons in real time on images with uneven distribution that can be efficiently run locally in the same mobile device used to pilot the drone?"

Like many others, the Castelldefels police patrol uses small and compact DJI commercial drones, because of their good quality–price ratio. According to the global market website Statista 2022, DJI's market share was 76% of units sold worldwide [4]. The main objective of our work is to add the functionality of counting people to this type of drone. Our experience is shared in the paper for those who are aiming at adding a similar functionality to their drones.

The work has an evident contribution beyond the state-of-the-art that is provided in three incremental steps:

- Leveraging AI capabilities into the piloting device of a drone. For the first time, we show that this can be performed without disturbing the critical flight functionality of the drone, and, thus, it does not affect the safety of the operation. Moreover,



the proposed solution avoids any additional weight on the drone, and, thus, it does not affect the endurance of the flight.

- A novel approach to speed up the execution of a deep learning detection model. Current methods rely on using GPU hardware or on reducing the neural network size. Adding GPU hardware has been applied on board the drone in previous works, but GPU is a high-power consumption device that affects the flight time of the drone. The neural network size-reduction methods (e.g., quantization, pruning or knowledge distillation methods) are known by their loss of accuracy. To our knowledge, the extensive use of CPU cores of the piloting hand-held device of the drone has not been demonstrated before in this context. This contribution permits the execution of large AI models, which allows to return accurate counts in less time in a hardware-limited device.
- The training of a new AI model to segment crowds in an uneven density image. This model fosters the smart combination of two AI algorithms and permits solving the persons count with higher accuracy than the algorithms separately.

The organisation of the rest of the paper is as follows: Section 2 presents the state-of-the-art of drone image processing methods; Section 3 shows the three progressive solutions we have developed to help law enforcement in their surveillance tasks; Section 4 presents the results of our three solutions; and Section 5 puts them into context with existing models, addresses conclusions, and sets out future trends of drone applications for law enforcement.

## 2. Previous Work

The absence of a pilot on-board in drones has partially been solved with one or more cameras on-board and the communication link to download the images captured in the air. Very early on, the capabilities of image processing arose to help the pilot in flying and maintaining safety. Image processing proposals have been published for landing area recognition [5], as support to an emergency landing operation [6], and as a detect-and-avoid on-board system [7].

High-definition cameras are also the most commonly used payload sensor, together with infrared cameras or LIDAR. The processing of images from the payload cameras can be used in missions such as building inspection (i.e., the identification of harmful asbestos slates [8] or the identification of damage to multiple steel surfaces from panorama images [9]), road traffic surveillance [10], and search and rescue [11]. Automatically labelling the images with significant information helps law enforcement agents to detect situations that need to be corrected or to find people that are lost. For the pandemic, research has demonstrated that drone images can help to detect and differentiate people with and without a face mask by using drone images [12,13], and also to measure social distancing during the pandemic [14]. Hammer et al. [15] propose a solution for detection and tracking of persons using LiDAR.

Additionally, drone image processing can add counting capabilities useful for capacity and accountability management. Works can be found that count cars [10], plants [16–19], or animals [20].

Most of the above works are developed on artificial intelligence (AI) algorithms. The use of AI in object detection is based on supervised learning and convolutional neural networks (CNN [21]. In the literature today, we can find many CNN proposals for object detection, such as the classical VGG-16 [22] with only 13 convolutional layers, to deeper networks, such as the Mask R-CNN two-stage detector [23] or YOLO one-stage detector (with an increasing number of layers in the new versions of the algorithm) [24]. Variants and combinations of known backbone CNN have been created to improve detection in specific situations. A very extensive and well-studied mission is the detection and geo-localization of humans lost in nature (search and rescue or SAR missions) [11,25]. The authors propose new networks, such as RFCCD, that combine three former networks.

In [26], a thorough study on CNN proposals applied to detect small objects shows that the state-of-the-art detectors obtain low accuracy (less than 40%) in these targets.

The detection of small objects, defined as less than $50 \times 50$ pixels, remains a challenging task. Problems mentioned are insufficient information captured by the feature detection layers, limited context information, the unbalanced ratio of background versus a small sparsely located object (with ratios from 100:1 to 1000:1), and insufficient positive examples for the training.

The small object detection problem becomes even worse for images with a dense crowd. When using a detection algorithm on these images the bounding box of a person overlaps with the neighbour's bounding box, complicating the computation of the loss function. For this reason, other methods based on Density Map estimation, such as CSRNet [27] and STNNet [3], are mostly used to count crowds. Additionally, using also deep CNN, these algorithms do not find the anchor box of each detection. Instead they train with density maps of the same size as the input images. Each image is annotated by adding one pixel in the centre of each object, generating a sparse binary mask. Then, a pseudo ground truth density map is created by blurring the ground truth, with a Gaussian smoothing process. The result is the truth density map used as training set.

A new approach is used by Wang et al. [28] in which the ground truth image is not blurred in advance to the training, but included in the loss function. Named as Distribution Matching, this algorithm uses the Optimal Transport function as a loss function to optimize the density map similarity with the binary annotated ground truth.

Luckily, the number of public datasets containing persons, open to train existing and new algorithms, is growing. The most popular is probably the Shanghai dataset [2], with a part-A with highly populated scenes and part-B with sparsely located persons. Increasing the persons density, we find the UCF-QNRF dataset [29] and the crowd dataset [1], the latter with image frames extracted from videos recorded at drone altitudes. Datasets, such as the SAR dataset [25], also contain images taken from drones, in particular, for search and rescue scenarios.

The public datasets and the challenges opened to the research community, such as the VisDrone 2021 Crowd Counting challenge, contributed to an impressive improvement in the crowd counting problem. Results for the VisDrone 2021 dataset, with high-altitude images with up to 403 persons and fewer than $15 \times 15$ pixels per person, have a mean average error of less than 20 persons (see Figure 2). All of the research works listed in the figure proposed the use of deep neural networks with new architectures, new loss functions, and new algorithms for training.

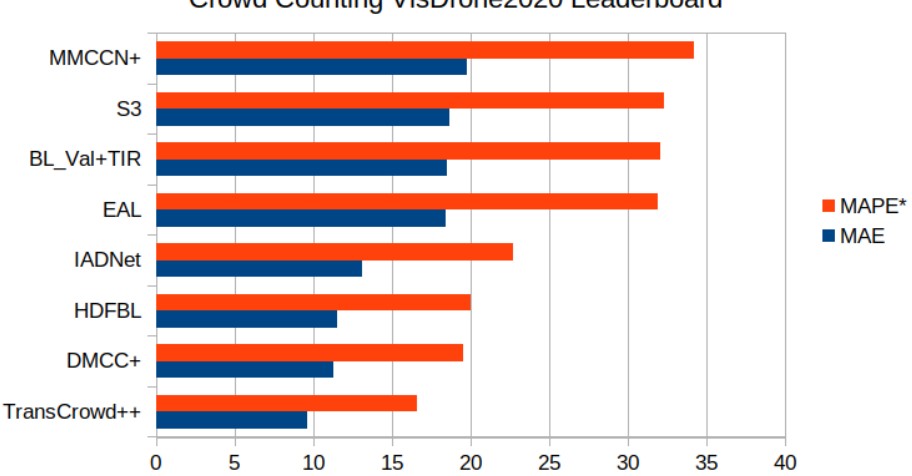

**Figure 2.** Leaderboard of the VisDrone 2021 Crowd Counting challenge [30]. MAE stands for Mean Absolute Error and *MAPE* for Mean Absolute Proportional Error. *MAPE* equation is given in Section 2 and for the leaderboard (*) it is calculated globally with the average persons of the full dataset [31].

The challenge still remains open for running these successful solutions on the small devices of the typical drone used by law enforcement agents. As Ref. [15] highlights,

it is very important retrieving results in real time and presents a novel approach based on panoramic point clouds. Unfortunately, LiDAR sensors are not yet available in many commercial drones, and not found in local police fleets. LiDAR sensors are also significantly big, in relation to small drones, and their weight and power consumption reduce the maximum flight time. The contributions presented in this paper are based on current drones' sensors and do not impact in the weight nor in the power consumption of the drone.

Advances in other areas of application, such as autonomous driving, and those interested in detecting traffic signals and pedestrians, have similar limitations. Techniques to reduce the size of the CNN can help to reduce memory requirements and energy consumption. Krishnamoorthi [32] provides a survey of the best methods for quantizing deep convolutional networks using visualisation tools. Li [33] provides a tool to prune the less-useful neurons of a CNN. Ad hoc implementations of most popular networks (i.e., YOLO-Tiny [34]) are also available. However, the consequence of reducing the size of a CNN is some loss of accuracy. The balance between requirements on quality and resources has to be carefully considered and tested.

### 3. Materials and Methods

Drones are also known by the name of unmanned aircraft systems (UAS). This plural denomination highlights very clearly that, in addition to the flying vehicle, a drone has a second important subsystem: the command and control (C&C) device in the hands of the pilot-in-command. Other typical names of the C&C are remote control, ground control station, or smart controller.

The way the DJI drones work is very similar to most drones. The unmanned aircraft has a computer on board that manages the flight control and the payload, while the C&C has a display to inform the pilot about the flight and offers buttons, sticks, rollers, selectors, and other input mechanisms to command the drone. In between, one communication channel transmits orders from ground to air and downloads the flight status. Typically, a separated communication channel exists for the payload transmissions.

The payload data have a much higher volume than the flight data. First-person view video transmission is usually active, but in low resolution (for instance, 0.150 Mb per frame at 30 fps), thus it consumes little bandwidth. Especially critical are the high-resolution images that current commercial drones can capture. Image size can be up to 8 to 12 Mb each. High-resolution images are typically held in the on-board memory card and transmitted to the ground only upon demand. To be able to select a specific image, the low-resolution footage of the images is downloaded automatically to the C&C.

Considering the above-mentioned characteristics of the functioning of drones, a three-step approach has been taken for counting the number of persons. We named the solution *POLO* in reference to its end-user's name (*POlicia LOcal*), but also because of the similar phonetics with YOLO, the algorithm used.

1.   POLO*tin*: the initial proof of concept that uses low-resolution image footprints and the state-of-the-art YOLOv3-Tiny [35] detection algorithm and trained model;
2.   POLO*par*: this is an extension of the previous solution able to process high-resolution images. Here, we trained a new model with our own set of police images to improve accuracy. The main contribution is that POLO*par* parallelises the detection algorithm in several threads, in order to improve the efficiency of models with good detection accuracy;
3.   POLO*seg*: this solution completes the full contribution by addressing the counting of highly congested scenes in parts of the image. The main difference with other approaches is that the highly congested parts of the image are segmented from the non-congested, using different methods for each part.

The first two have been fully developed as Android Apps. They are tested with runs in the command&control of a DJI Mavic2 drone, a model that is often used by law enforcement crews. Both a DJI Smart Controller and a commercial off-the-shelf rugged tablet have been tested satisfactorily. To access the drone hardware and to create the user interfaces we used

the DJI MOBILE and UX software development kits (SDK), and for building the neural network and loading the model weights we use the OpenCV library. The POLO*seg* was set as a proof of concept on how to improve the counting on challenging images with uneven density crowds.

### 3.1. POLOtin

This first contribution has three primary objectives: to use an efficient object detection model from the state-of-the-art, to embed it in an Android application, customized to detect persons on drone images, and to propose a human interface to show on the C&C the total count to the agent commanding the drone.

For the model, we select the lightweight version of consolidated object detector YOLOv3 [24], named "YOLOv3-Tiny" [35]. This is a lightened version of the original algorithm as a result of reducing the layers from 106 to 24, being the convolutional reduction from 53 down to 7 and interleaving them with 6 max-pooling layers. The size of the model decreased from $63,882$ values to $15,210$, 76.2% smaller. Despite the size reduction, the detection is performed on the same basis as YOLOv3 but just applying two prediction scales: $13 \times 13$ and $26 \times 26$.

The advantage of the tiny network is its high-speed prediction and small footprint, being ideal for mobile devices where resources, such as memory usage, are key parameters when developing the Android application.

Launching the POLO*tin* it initially shows the flight interface. This is a very similar interface to the DJI Pilot App, from which the drone can be managed and pictures/videos can be taken. It also has a second interface where the detection model is executed. Figure 3 shows a capture of it.

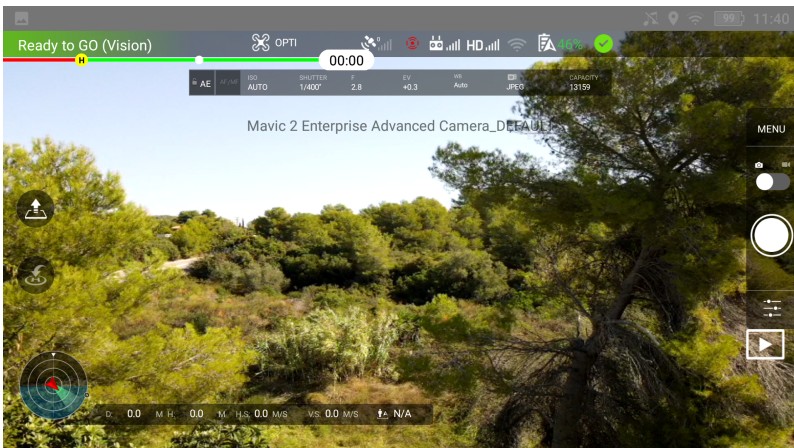

**Figure 3.** Flight interface of POLO*tin*.

The execution flow for the detection is as follows: first, we look for the footprints of the images taken during the flight and show them to the user. When the user selects one of the images then we create the neural network, update its weights with the tiny model and run the detection algorithm on the image. Since this is a small footprint of the real image the process is fast (less than 2 s). The model is able to detect 80 different objects, but we filter only those objects classified as person, show them in bounding boxes and provide the total count. Section 4 shows the results on a test flight performed with students.

### 3.2. POLOpar

After demonstrating with POLO*tin* that it is possible to use state-of-the-art AI models in DJI drones, we progress with further functionalities. With POLO*par*, we aim at processing high-resolution images, downloaded from the drone upon the pilot's request, and applying deep AI models with high detection accuracy.

A new human interface allows the selection of any image, as shown in Figure 4. On the right, we now observe two new image galleries, in addition to the previews of the images in

the drone. This one is now named as the Remote Gallery, and, as before, it shows the images in the SD card on the drone. These are the low-resolution images processed in POLO*tin*. By selecting one of them, the high-resolution image is downloaded to the C&C disk, inside the Downloaded Gallery. Figure 4 shows some images downloaded in previous runs. Finally, when selecting an image from the Downloaded Gallery, the detection algorithm runs automatically and the result is shown (see the left-hand side of Figure 4). In addition, the result is stored as a new file in the Processed Images Gallery. After processing, the image contains the detection bounding boxes and the total count at the bottom right of the image.

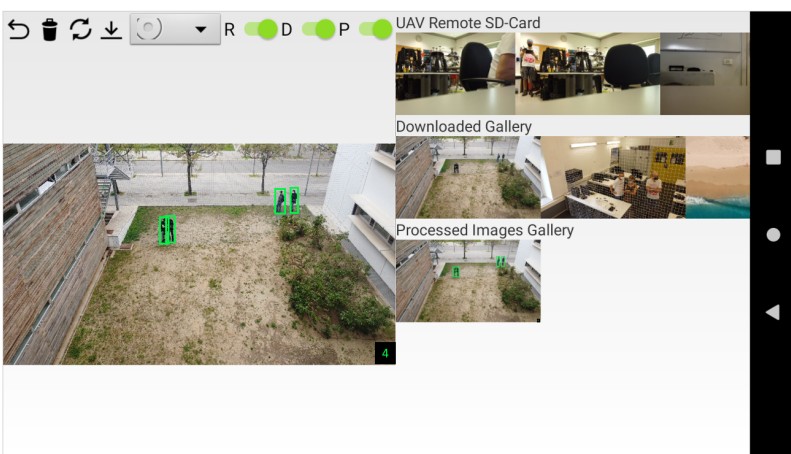

**Figure 4.** POLO*tin* human interface design and developed functionalities.

We also use the YOLOv3-Tiny network in POLO*par*. The input layer of the model is $416 \times 416$ px, while the images downloaded from the drone are $4000 \times 2250$ px. In most police images the average size of a person to be detected is around $8 \times 20$ px. If we scale the image down to the size of the model we will lose most pixels and will highly compromise the performance of the model. Thus, we decided to train the new model with the current pixel size of the bounding boxes. For this the images need to be cropped into sub-images, as shown in Figure 5, instead of down-scaling them.

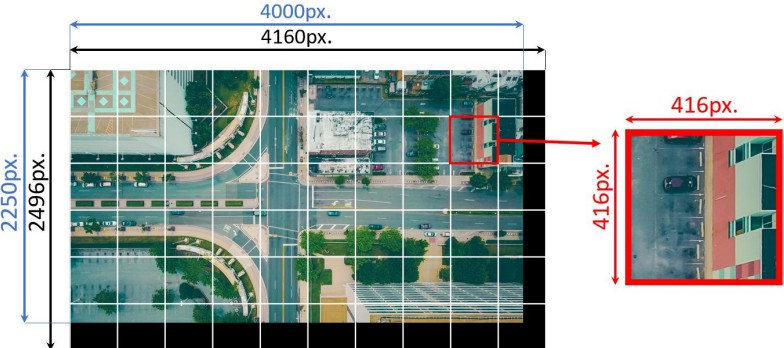

**Figure 5.** Example of the high-resolution image cropping that preserves the pixels per person ratio.

For the model training, we manually labelled the police images. Then the bounding boxes were converted from the absolute coordinates system of the high-resolution image to the relative coordinate system of the cropped images, and vice versa during the inference. This is performed automatically and supported with text files and a specific folder structure.

After cropping we had 7776 sub-images for the training. This was conducted in a cluster with 2 Intel Xeon 4210R CPUs, 8 2080Ti GPUs with 11 GB of VRAM each, 128 GB of RAM memory, and 2 TB of disk storage. It took 11 h to complete a training of 40,000 iterations. With the help of transfer learning, the convergence of training was fast with a stable loss after only 2000 iterations and a final loss of 0.353.

The POLO*par* uses the trained model for the person detection. The detection process works as follows. First the selected image is cropped to 16 images of size 416 × 416 px. Figure 5 shows that the division is not exact and padding is needed in both width and height dimensions. Each cropped image is then sent to the CNN model and the bounding boxes of the detection are generated. Once the 16 images are processed, the high-resolution image is reconstructed and the bounding boxes converted to the absolute pixel reference. The full process is very CPU/GPU consuming and the elapsed time is considerably long (see more details in Section 4).

To improve the detection time, we use multi-threading. Multi-threading is the ability of the CPU to provide multiple threads of concurrent execution. When this CPU has several cores, the concurrent execution is completed in parallel and the elapsed time can be reduced. Fortunately, modern handheld devices come with multi-core processors that support this feature. In consequence, the POLO*par* App distributes the cropped images among parallel threads. Determining the best number of threads to use is very important. It depends on two main hardware specifications: the number of CPU cores and memory size.

Section 4 details the hardware characteristics of our device and the execution time of the process for different multi-threading configurations. The final configuration, with 6 threads, 1 per row of cropped images, is shown in Figure 6.

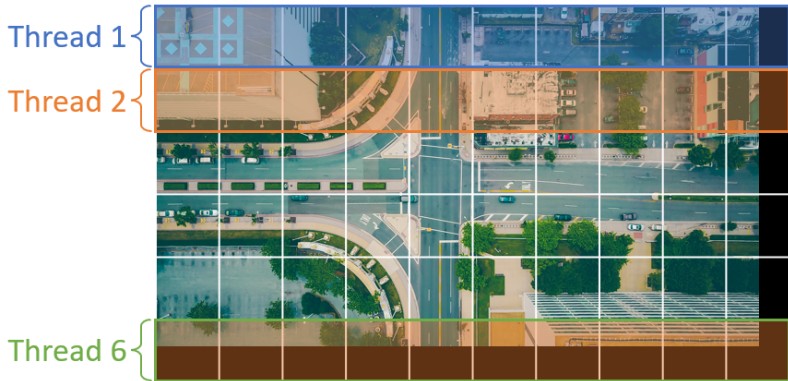

**Figure 6.** Parallel execution with 6 threads.

### 3.3. POLOseg

POLO*seg* is developed focusing on the motivating image of Figure 1. This is a proof of concept to address more challenging images with dense crowds. It is developed in Google Colab, with one GPU and CUDA activated for parallel execution. The network implementation is performed with Keras and Tensorflow [21]. The proof of concept can be easily integrated in the previous App by providing the police officer in command with a new option in the selection interface to apply the crowd pre-processing to the image. In the POLO*seg* concept, the image is segmented and the crowd processed separately from the remaining, sparser parts of the image. Figure 7 shows the steps in the pre-processing of crowds.

The segmentation of the crowd is based on Mask R-CNN [23], an extension of Faster R-CNN framework. We train a new model able to detect and segment any object with any shape. First, we manually label the crowds on 600 images, some coming from the Shanghai dataset in which crowds can be separated from the background, and others from our own source (local police and pictures taken from drones mostly in Castelldefels). Some images were also extracted from videos. The training process took around 10 h.

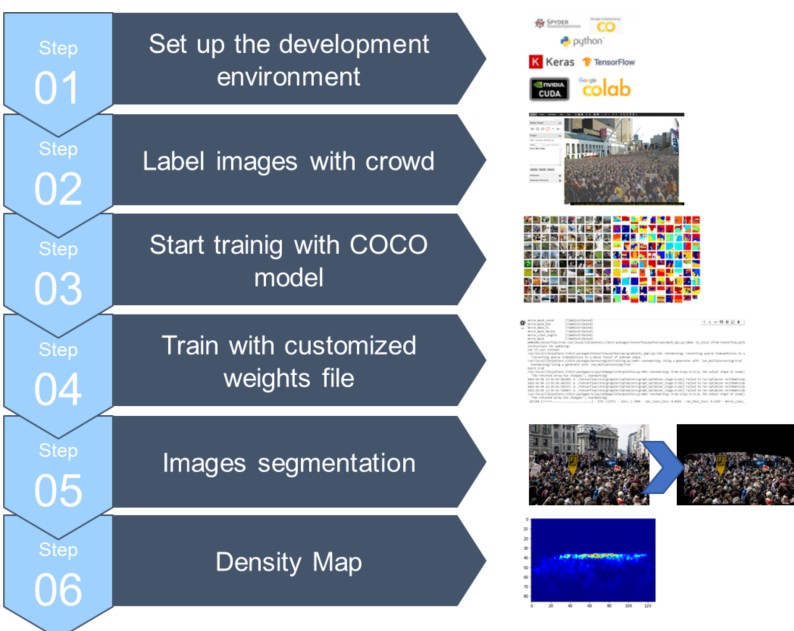

**Figure 7.** Steps of the development process of POLO*seg*.

Table 1 shows the distribution of the images used for training and for testing.

**Table 1.** Breakdown of datasets and images used during the training process.

| Dataset | Total | Training | Testing |
|---|---|---|---|
| Shanghai [2] | 473 | 434 | 39 |
| Castelldefels | 171 | 154 | 17 |
| UCF-QNRF [29] | 100 | - | 100 |
| **TOTAL** | **744** | **588** | **156** |

Figure 8 shows the crowd segmentation process, corresponding to steps 02 to 05 in Figure 7. The pipeline used for the training process is shown at the top of the Figure. The training uses as input the model trained with MS-COCO and loops over our set of crowd-labelled images to adapt the model to our needs. The iterative process is stopped once we obtain a segmentation with an intersection over union of 75%. Once the training loop finishes, the validated model is applied to the fresh test images (shown in the bottom left of the image). In the end, the part of the image that contains the crowd is segmented (the 1 in the blue dot).

For each input image, the program generates two sub-images: one renamed as *segmented* and the other as *negative*, with both keeping the original image name at the beginning. The *segmented* image is the image containing only the crowd and the *negative* is the opposite, the remaining part of the image that is not a crowd. This last sub-image is processed as in POLO*tin* and POLO*par* to detect isolated persons.

For the segmented image containing crowds, a second step is added in which the density map is generated using CSRNet [27] (step 6 of Figure 7) after the input image is resized down to 1024 × 1024. CSRNet uses VGG-16 [22] as the front-end to extract features of the input image because of its strong transfer learning ability and its capability to easily concatenate the back-end for density map generation. CSRNet provides two additional functionalities: to graphically represent the density map of congested scenes, and to count the number of objects in the density map. We use the model pre-trained with the Shanghai dataset. Figure 9 shows this part of the process. The crowd counting can run in parallel with the individual person detector algorithm of the POLO*par* version. Both counts are added at the end to obtain the global value.

CSRNet is a deep learning method able to give a count estimation figure of an input image and to create its density where the crowd distribution is shown. The density map plays a key role in understanding the results as one can see in a graphic way the congested area in the original image and check the quality of counting. Like most deep learning networks, it is composed of a front-end and a back-end. The front-end is a Convolutional Neural Network for 2D feature extraction. The back-end is a Dilated Neural Network which uses dilated kernels to deliver larger reception fields and to replace pooling operations.

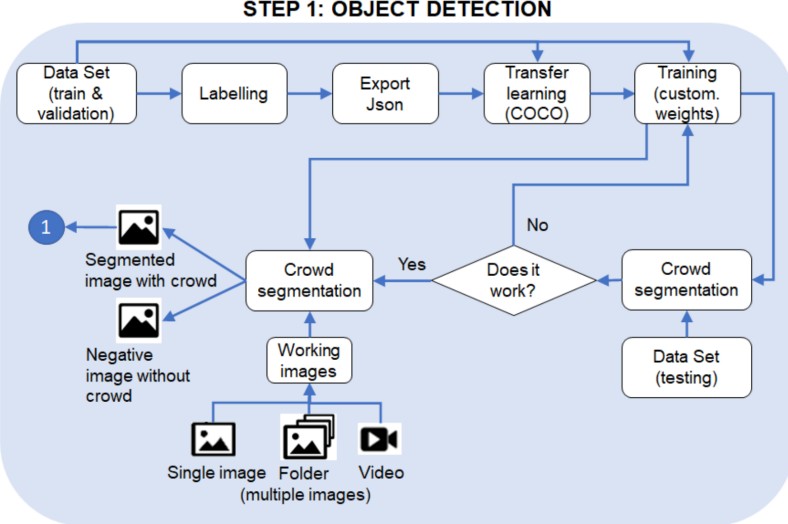

**Figure 8.** Crowd segmentation: training and testing processes.

CSRNet also provides a model trained with four crowd datasets: ShanghaiTech, UCF-CC50, WorldEXPO'10, and UCSD, and applying augmentation techniques.

It uses small-size convolution filters (like $3 \times 3$) in all layers. The front-end has the first 10 layers of VGG-16 [22] and the back-end has dilated convolution layers to enlarge receptive fields and extract deeper features without losing resolutions (pooling layers are not used).

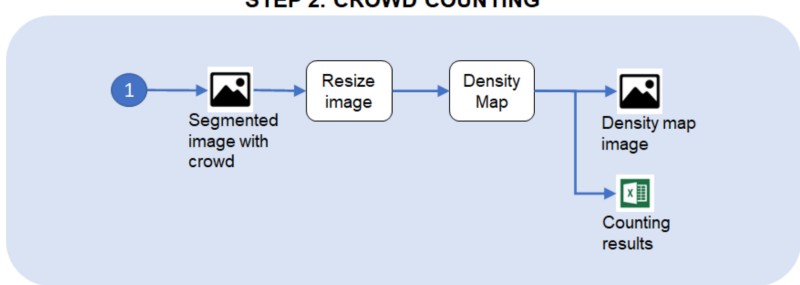

**Figure 9.** Crowd counting using Density Map.

The results of the segmentation and of the crowd counting are presented in Section 4.

## 4. Results

This section presents the results of the three algorithms. First a qualitative view of the processed image, displayed to the end users, is given. Then quantitative results are given for the count error (details in next subsection) and for the execution time.

### 4.1. Metrics Used for the Qualitative Evaluation

To evaluate the results of the person count we use the mean average proportional error (*MAPE*) metric. Although the mean average error (MAE) metric is used more extensively in state-of-the-art works, it has the same scale as the data being measured and, thus, it is a

scale-dependent accuracy measure. Therefore, MAE cannot be used to make comparisons between series using different scales. *MAPE* is a well-known metric in the literature [31] and more understandable for the end-users. The *MAPE* metric is very similar to the MAE (mean average error) but puts the absolute error value in context with respect to the total count of each image. For test datasets, such as the ones we use, where the number of people in each image is very different, we believe this metric is the most appropriate.

Numerically, the *MAPE* is defined as in Equation (1):

$$MAPE(\%) = 1/n * \sum_{t=1}^{n} |\frac{trueCount_t - predCount_t}{trueCount_t}| \qquad (1)$$

where $n$ is the number of images, $trueCount_t$ is the ground truth value for image $t$ and $predCount_t$ is the resulting count value of the counting algorithm.

In object detection and in segmentation the results include a bounding box or a polygon, respectively, where the object is predicted to be. The quality of the detection/segmentation is given by the overlap of the areas of detection and truth. When the overlap is too small or null, the detection/segmentation is considered incorrect. To measure the overlap we use the typical metric of *IoU* (intersection over union), defined as in Equation (2):

$$IoU(\%) = \frac{Area\_of\_overlap}{Area\_of\_union} \qquad (2)$$

where areas are measured in pixels. The two areas of comparison in *Area_of_overlap* and *Area_of_union* are the resulting areas of the automatic segmentation process and the manual segmentation area considered as ground truth. An *IoU* value of 1 represents a perfect segmentation, while smaller values are less accurate results.

A threshold value of the *IoU* has to be defined to decide between a good and a bad prediction. For detection (POLO*tin* and POLO*par*) we set the threshold to 0.25 and for segmentation we set it to 0.75. When the calculated *IoU* is below the threshold the detection/segmentation is considered incorrect.

### 4.2. POLOtin Results

The YOLOv3-Tiny network, with the ImageNet trained model, was successfully integrated in the drone ground station as POLO*tin* and tested for the detection of persons during flight. A screenshot with the result of the detection and of the counting can be seen in Figure 10. Observe that the individuals detected are shown to the law enforcement agent inside red bounding boxes, and the total count is also visible in the top blue box.

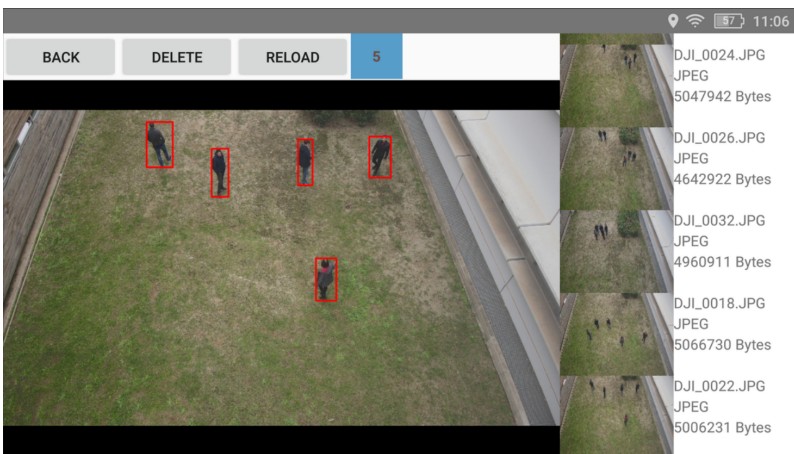

**Figure 10.** Person counting using POLO*tin* works well for low-altitude and separated targets.

The figure also shows on the right the information about all the images that are available in the drone. These images are just thumbnails and have a size of $99 \times 99$ pixels,

while their actual size in full resolution at the drone disk is around 5 Mb. For an efficient execution of the tiny model the image downloaded and used in the detection algorithm has a size of $960 \times 540$ pixels (around 500 Kb), one order of magnitude smaller than the full resolution image.

The use of medium resolution images allows the full process to be fast. From the selection of the image to the display of the detection results on the *C&C*, it takes only 1–2 s. The time depends on the quality of the communication link, but also on the parallel actions that have been executed with the payload. For instance, the drone is able to capture video at the same time it takes images, but then the elapsed time seen by the user is longer.

We ran the POLO*tin* detection algorithm for 70 images. From these, 55 images are similar to the one in Figure 10, test images taken at the school, and 15 images are from another dataset. This second dataset has images of Barcelona beaches in summer, with a much more challenging detection than the one shown in Figure 10. Figure 11 shows the comparison between the actual number of persons and the predicted one, and Figure 12 the characterization of the images in terms of the average size of a person in pixels. Notice the important differences between the two sets of images: for the set of images taken at the school (the first 55 images) and the set of images taken at the beach (the last 15 images).

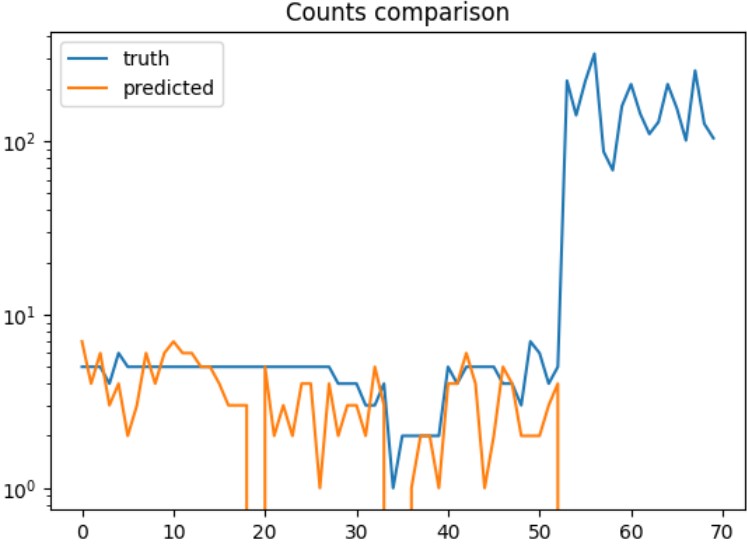

**Figure 11.** Person count: predicted vs. truth.

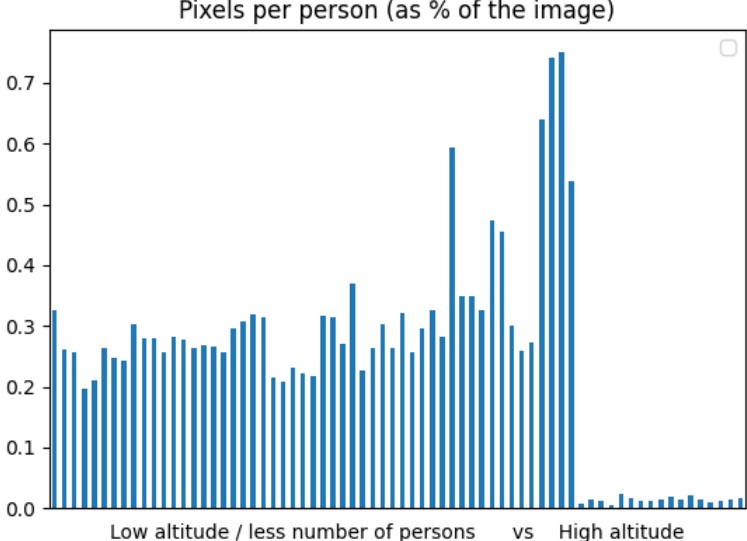

**Figure 12.** Image characterization (pixels per person, as a percentage).

For the first 55 images the average resolution of each person is around 0.2–0.3% of the pixels of the image (bounding boxes of 300–500 pixels). For these images the count error (1–2) is acceptable, as expected for the state-of-the-art model used.

For the second set the resolution of a person on average is less than 0.02% of the image (bounding boxes of 10–30 pixels). Observe that the state-of-the-art tiny prediction model is unable to detect any persons when the drone is flying at high altitudes as is the case in the beach images. The *MAPE* of this set is 1 (100% of error) for all images.

The predictions of the person count, compared with the truth count, using integrated state-of-the-art methods integrated in the drone system can be considered as a good starting point for the school images. More detailed results for these images are shown in Figure 13, with the *MAPE* averaged by ground truth values. Globally the *MAPE* for all the low-altitude images resulted in a 35% deviation from the truth count (see average line in the figure). In seven images, the count is perfect, while in three of them there is a detection when no person was in view (left bar). Most of the images had five persons in view, for which the *MAPE* is even below the average.

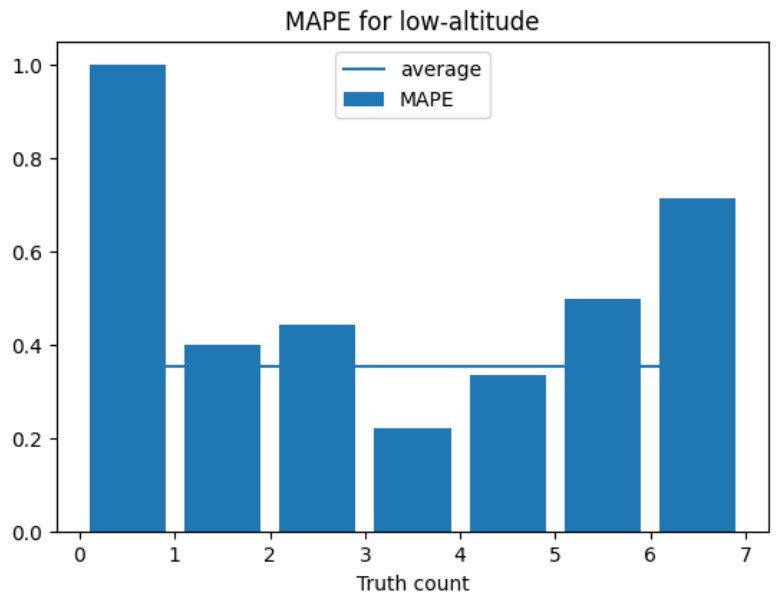

**Figure 13.** Mean average proportional error (*MAPE*).

### 4.3. POLOpar Results

A new trained model was used for obtaining the results of POLO*par*. The training was conducted with 7776 cropped images (both for training and validation). Using the 4-GPU cluster it took 11 h to complete a training of 40,000 iterations. The final average loss obtained was 0.353230, defining loss as the *IoU* function.

The new model, also using the YOLOv3-Tiny network, has 12,200 weights, 2.1% smaller than the POLO*tin* model given that the number of classes is reduced from 80 to 1. This is two orders of magnitude less than the YOLOv3 model which has 690,000 weights, and three detection layers instead of two of the tiny model.

The main results are given with the following two figures: Figure 14 shows the persons detected in the challenging picture presented in Figure 1; Figure 15 shows the execution time of the detection of an image in POLO*par* on a Tripltek 7in Pro rugged tablet, with a Snapdragon 845 (octa-core CPU) with 6 GB of memory.

Looking at Figure 14, we observe that the quality of the person detection in the marathon image is very good for the persons that are almost fully visible and well separated from others. The total count, 236 persons, is very accurate, with only 16 persons missing (*MAPE* is 6.78%). Nevertheless, the compact crowd is clearly not detected by our new model. Further executions show that this area is not detected with any of the available

existing (no-tiny) models of YOLO and that it needs a different algorithm, such as the Density Map presented in Section 3.3.

The results on the other hand become worse with the school images (*MAPE* is 100%). If in App $POLO_{tin}$ the 4–6 persons were well detected given their high pixel density, the cropping is clearly unnecessary and inconvenient. For this reason, $POLO_{par}$ adds the option of applying different levels of cropping, with no cropping being the best option for these images.

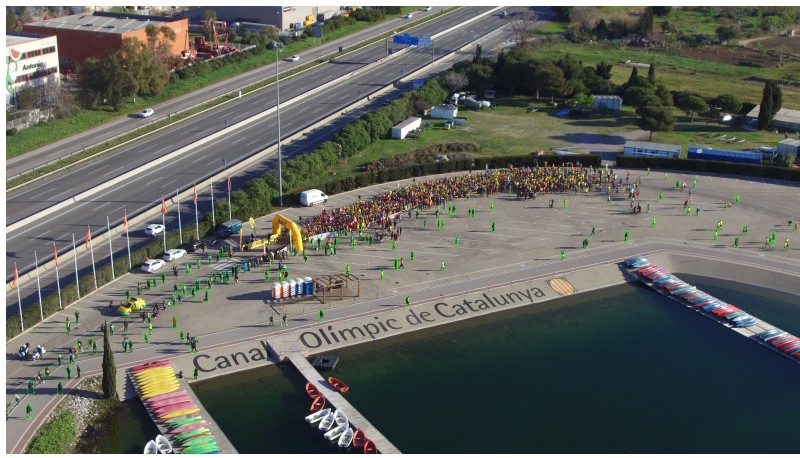

**Figure 14.** Visual result of the motivating image. The count is 236, that correctly the detects persons outside in the runners' area.

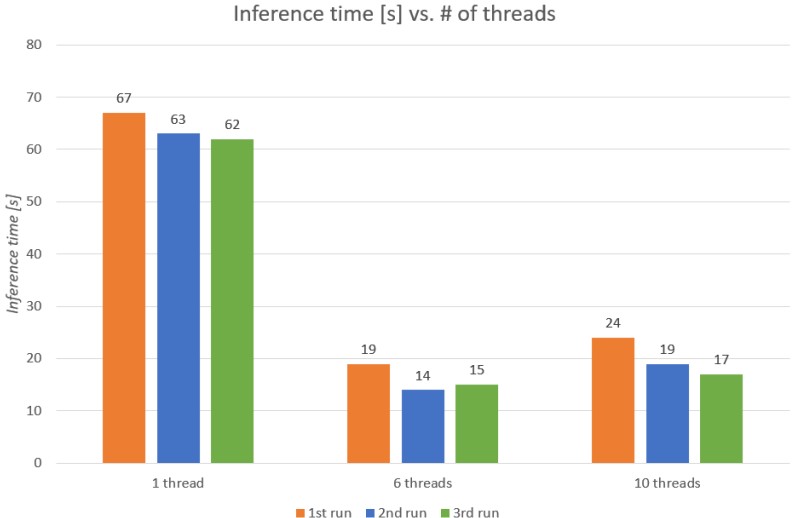

**Figure 15.** POLO*par* inference time depending on the number of threads.

Results on execution time are given in Figure 15. Execution times of the full process, including the cropping and the parallel execution of the detection, are shown for 1, 6, and 10 threads, and for 3 consecutive runs. The first run is always slower because the CNN has to be loaded into the memory of the *C&C*. It is observed that applying the POLO*par* model as in the state-of-the-art solution (with 1-thread execution) raises the execution time from the 2 s of model POLO*tin* to 62–67 s (see 1-thread bars in Figure 15). The slowdown is due to the execution of the detection to the 60 images resulting from the cropping process of a full resolution image.

When applying our parallel solution, with the execution of the detection distributed across the CPUs of the ground device, the optimum is found for 6 threads, obtaining an execution time of 14–19 s.

Other parallelisation strategies, such as processing the sub-images of each column of the cropped array in separate thread (10 threads in this case), result in less efficiency. The reason is that the number of threads exceeds by two the number of cores of the CPU. By assigning a row of 10 images to each of the 6 threads we find a good combination in which there are still CPU cores available to control the rest of the application (i.e., the flight functionalities) and for the operating system.

The prediction numbers of the POLO*par* model can be compared with the ground truth in Figure 16 for the same subset of 70 images as before. Two tests of the model are shown, one with no image cropping (named as *NoCrop*) and a second one with 416 × 416 cropping (named as *Crop*416). The execution of *NoCrop* is much faster (1–2 s, as in POLO*tin*) but results for the high-altitude images are also very bad (*MAPE* close to 100%). We observe how the same trained model can obtain better results by means of cropping. Cropping allows it to work with the pixel resolution available at the high-resolution original image. Additionally, parallelisation allows to keep the prediction time within the requirements of the end-user.

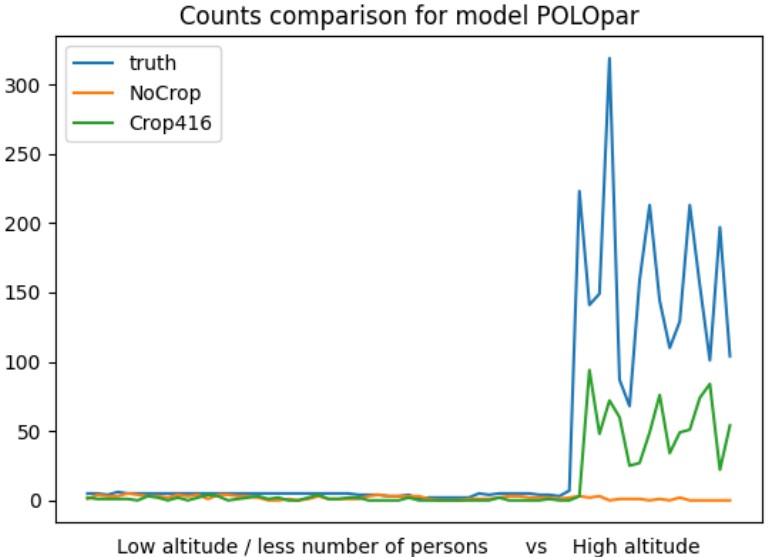

**Figure 16.** POLO*par* prediction results.

Looking further at the accuracy of the prediction for the high-altitude images of the Barcelona beaches, Figure 17 shows the average *MAPE* of several runs using our trained POLO*par* model after the full training from the police images, and compares it with the state-of-the-art MS COCO trained models for YOLOv3 and for YOLOv3-Tiny. Notice that YOLO-V3 uses a CNN with 106 layers, while POLO*par* and YOLOv3-Tiny models have only 23 layers.

We observe that the best results are for YOLOv3, the deeper CNN model. Still this model benefits from the proposed image cropping process to obtain good results.

Comparing the detection quality of the two small size models: the state-of-the-art YOLOv3-Tiny model and our model, we observe that the improvement is almost 40% when cropping is applied.

Our model, POLO*par*, is still clearly missing a significant percentage of the persons, with a *MAPE* of 67.6% for a dataset of high-altitude images not seen during the training. For the local police images, the error was 35.3%, which is still high for the expectations of the end users. There is clearly an important trade-off between the quality and efficiency of the detection.

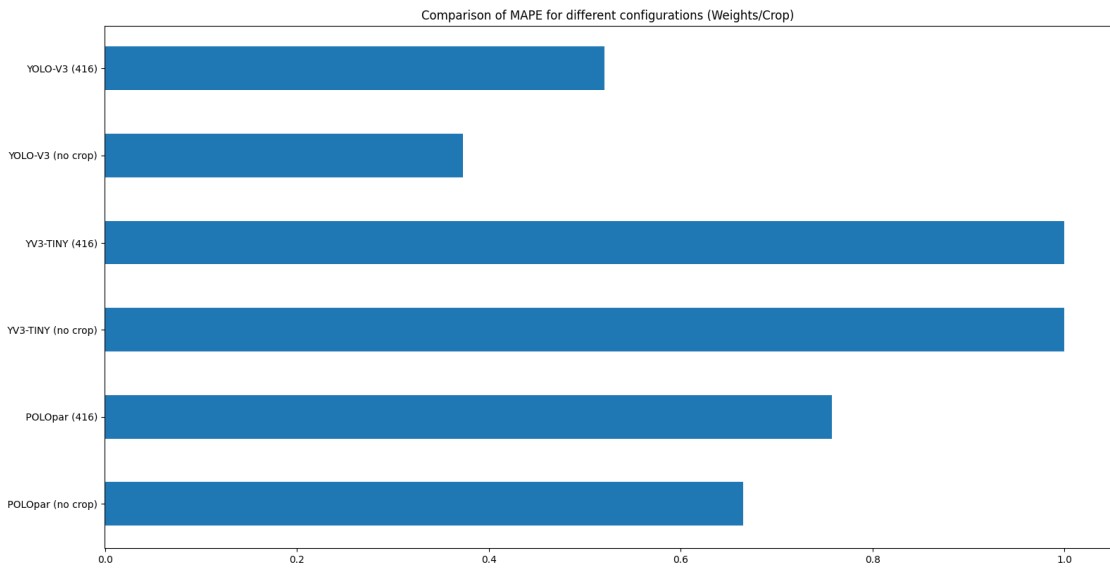

**Figure 17.** POLO*par MAPE* results for high-altitude images.

### 4.4. POLOseg Results

The results of the crowd segmentation and density map counting processes in the POLO*seg* application are shown in Figure 18 for an example image. On the left, we observe the original image, in the middle the result of the crowd segmentation, and on the right the density map and the total person count of this part of the image.

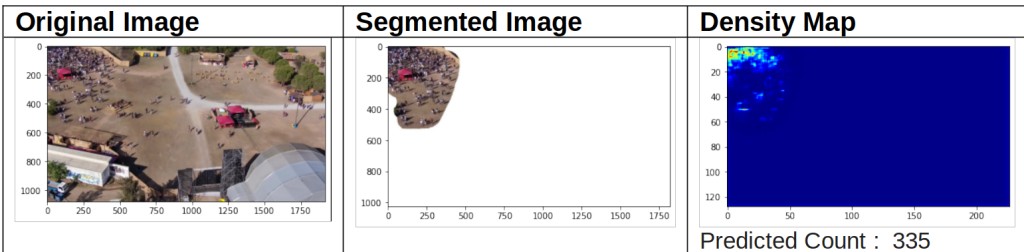

**Figure 18.** Person counting in 2 steps (Source: MARS Intelligence, permission granted).

More results of the segmentation can be seen in Figure 19, which also includes the motivating scene of Figure 1. The segmented images show that the detection of crowds has been mostly successful, and the background is eliminated for a better density map generation.

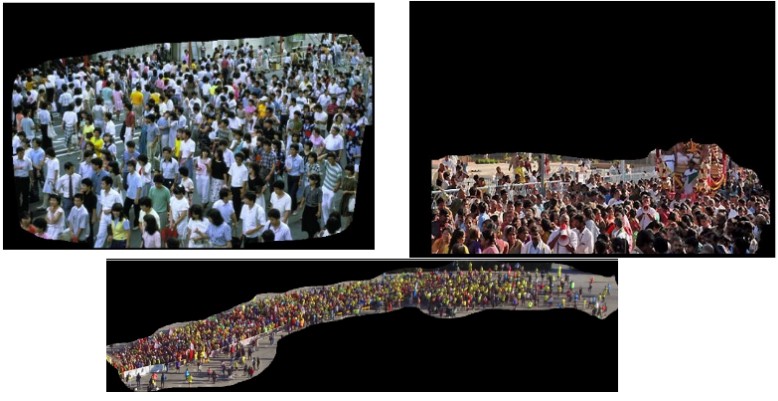

**Figure 19.** Results of the segmentation for three dataset images.

The full results of the segmentation are given in Table 2. On average, the images had around 60% of their pixels segmented as a crowd. The table shows the number and percentage of the test images for which the segmentation was considered correct. In terms of *IoU* a value of 75% or more corresponds to a correct segmentation. Since the test dataset was very successful, we also added some other images from our own dataset. Surprisingly, the Castelldefels dataset had 6 images that were not correctly segmented, where the algorithm failed in detecting any crowd. Most of these images are from the police, and the altitude was much higher than in the rest of the dataset images. This highlights the important effect the altitude has in the detection algorithms.

**Table 2.** Results of the segmentation of 56 images.

| Dataset | Total Images | Correct | Percentage |
|---------|--------------|---------|------------|
| UCF-QNRF [29] | 100 | 100 | 100% |
| Shanghai [2] | 39 | 39 | 100% |
| Castelldefels | 17 | 11 | 65% |
| **TOTAL** | **156** | **150** | **96%** |

However, counting results after crowd segmentation showed no significant improvement of the metrics (less than 1% improvement for the *MAPE*) with the type of images used in the test (images taken from the Shanghai dataset, with an average of 60.02% of image pixels representing highly congested areas). Anyway, segmentation is still useful, as the Density Map method is not able to count isolated persons. Additionally, background objects (trees, buildings, sky, etc.) are adding noise to the count that is better to eliminate.

A non-seen set of 100 images was taken from the UCF-QNRF dataset [29], a public dataset with high-resolution images of crowds (average image size of 3926 Kb), to be used as the POLO*seg* test set. Only the aerial images were selected, the average image size of which was 144 Kb.

In addition to the lower rate of success in segmenting high-altitude images, such as the ones in the Castelldefels dataset, the execution time raised a very important drawback when testing the POLO*seg* algorithm on this test set. Running in the cloud, the segmentation algorithm for the UCF-QNRF dataset takes 27.46 s per image, on average. This is a very long delay for the expectations of the police officers. Compared with the average processing time of the Density Map counting algorithm (3.52 s per image, on average) this is, one order of magnitude higher. This is the reason why we decided to first evaluate the improvement in count accuracy obtained by the segmentation before we integrating it to the embedded *C&C* device.

The *MAPE* of the person count was calculated for the segmented images and compared with the non-segmented image. The ground-truth values of the counts had to be recalculated for the segmented images, since some of the persons outside of the crowd were not present in the scene any more. The results help to assess the benefits given by the crowd segmentation intermediate step.

Looking at Figure 20, we can observe in bars the individual values of the counting error, as a function of the crows count, and in dashed lines the trends of these errors. We observe that the average of the trend lines is between 20% and 30%, with no relevant differences between segmented and non-segmented.

As the number of people in one image (Figure 20) increases, the error is lower for non-segmented images. This is as expected, since the algorithm used was oriented towards very congested scenes. On the contrary, the segmentation converts all images into very-congested, and, thus, the *MAPE* trend line is more stable. In the middle of the plot (the number of persons between 500 and 1000) the segmentation shows some benefits, although they are not very significant. It is worth noticing that most of the images are within this range.

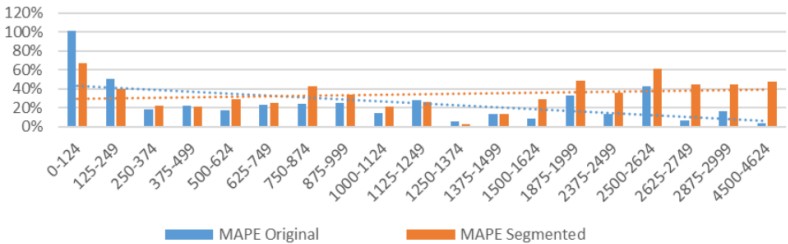

**Figure 20.** *MAPE* as a function of the crowd count (in orange with segmentation and in blue without segmentation).

The *MAPE* is shown in relation to the image resolution in the plot of Figure 21. This is a noisy plot in which no clear trends can be found. We can conclude that resolution is not a significant feature for crowd counting based on density map, and also that the segmentation of the crowd does not show a clear improvement over non-segmented (or original) results. Figure 22 combines the two features above in a new feature: pixel density, defined as the pixels per person of an image. The trends of this plot show that the error increases as the pixels per person increase. Again, this is a reasonable result for the algorithm used. *Small* people, in terms of pixels, are better detected than *big* persons. We may find that a reasonable limit is around 7000 pixels, which is close to a bounding box of $80 \times 80$ pixels on average. Segmentation is found to be helpful for these density levels.

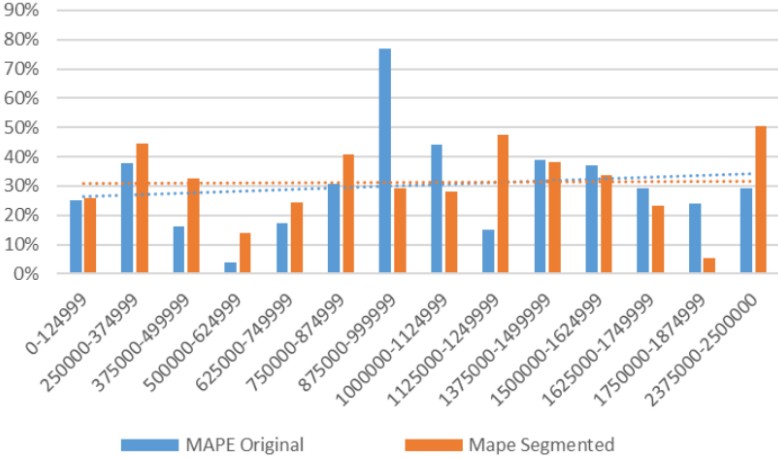

**Figure 21.** *MAPE* as a function of the image resolution (in orange with segmentation and in blue without segmentation).

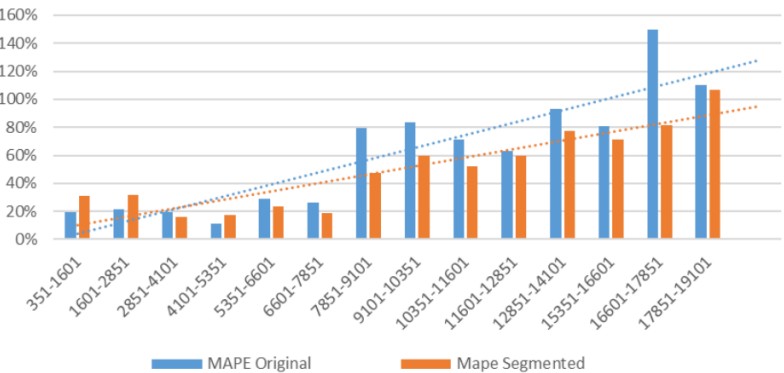

**Figure 22.** *MAPE* as a function of the density (in orange with segmentation and in blue without segmentation).

In the last plot (Figure 23), the error as a function of the percentage of the image that contains the crowd is shown. Highly congested scenes are typically close to 100%. For this

reason the trend line of the original algorithm reduces the error as the congestion increases. With the segmentation the trend is more stable.

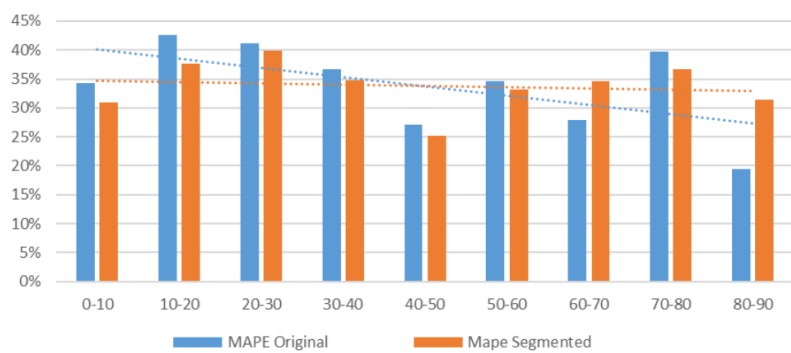

**Figure 23.** *MAPE* as a function of percentage of area segmented (in orange with segmentation and in blue without segmentation).

The following Table 3 shows a summary of the results of the three models and their main contributions:

**Table 3.** Summary of contributions

| Name | Model | Dataset | MAPE | Time (s) | Main Contributions |
|---|---|---|---|---|---|
| POLO*tin* | Pre-trained YOLO-Tiny | School (low-resolution) | 35% | 1–2 | Integration of state-of-the-art AI models in drone C&C |
| POLO*par* (with 6 threads) | New trained YOLO-Tiny | Police (high-resolution) | 35.3% | 14–19 | New model + Parallelisation strategy |
| POLO*seg* | New trained Segmentation + Density Map | UCF-QNRF (high-resolution) | 20–30% | 27.46 + 3.52 | Segmentation Strategy + new model |

## 5. Discussion

This work shows that it is possible to efficiently integrate state-of-the-art artificial intelligence models in a drone as new functionalities, embedded in the *C&C* device used to pilot the drone. In situations such as the COVID-19 pandemic, where strict control of capacity was needed, or other more common situations where large areas need to be kept within capacity limits, we have shown the possibility to easily add existing and improved deep learning detection models in the *C&C* of the drone for helping law enforcement agents.

With the proposed methodology, the way the agents manage the drone can be kept as usual, and with just one click any image onboard the drone can be downloaded immediately. Then, the image can be easily processed to count persons and the results can be presented to the pilot-in-command in less than 20 s thanks to the proposed parallelisation. Models and cropping levels can be selected so the agent can change the configuration when the results are not confident enough. As a rule of thumb, the higher the flight altitude, the deeper level of cropping is needed. Future work can be done to help to automatically select them.

Two model training processes have been executed. The first one allowed us to create an ad hoc AI model that improves the individual person detection of the images of the police dataset. This model can still be improved for other datasets of drone images, since count errors work for low-altitude images (*MAPE* from 20% to 30%), but are still too high (around 65%) in high-altitude images. However, the available public datasets do not contain images with the characteristics of the police drone flights. A perfect dataset shall be taken from altitudes between 60 m to 120 m, and very importantly, with some slant camera view. The task of labelling such images is huge but it is the confidential nature of the images that makes this task more difficult to achieve. Again, future work can be done in this topic.

The second training was conducted to create a model able to separate compact crowds from other parts of the image with more (individually distinguishable) persons. The segmentation results of this model were very good (96% of correct segmentation for 3 different datasets).

For the motivating image of the Mediterranean marathon (see Figure 1), the segmentation into two sub-images, followed by the execution of the object counting or crowd counting algorithms to the different sub-parts, was shown to be a good solution. It provided the best result, with a *MAPE* of only 0.14% (ground truth of 729 vs. combined result of 730). Several configuration alternatives were tried for this image, with errors varying from 8.64% to 77.64%. We checked alternative options such as: original high-resolution versus low-resolution, and different levels of cropping (original full-image versus cropped-image versus different crop sizes $416 \times 416$ pixels, $640 \times 640$ pixels). For the count of the crowd sub-image, the high-resolution and the full image seen at once gives better results than cropping or decreasing the resolution.

Although our methodology of segmenting the drone image works very well for the motivating example, when experimenting with the full test set the results show that the high computing-power demand of the segmentation algorithm does not justify its scarce benefits on the quality of the total count number. We believe, in agreement with the end users, that the best alternative is to leave to the pilot-in-command the decision about which algorithm is best for each scene. Density map is an algorithm oriented to highly congested scenes and provides a good count for them. According to our results, the ideal characteristics are given for high-resolution images with 250 to 1000 people, in which each person occupies around 1000 pixels, but never more than 8000 pixels, and the percentage of pixels with crowds is more than 80% of the area.

For future work, we plan to extend the deep learning models for other safety-critical situations. These will focus on vehicles, to monitor the traffic flow and detect jams, dangerous driving, etc. Other applications can include vegetation control in urban areas, safe monitoring from the air of an agent working on the ground in a dangerous situation, and the detection of illegal dumping of garbage or contaminants.

With the development of hand-held devices, with embedded GPUs and a high amount of RAM memory, the used AI models can progressively transition to deeper networks. This will, of course, improve the current *MAPE* results and lower the errors to make counts more and more reliable in real-time. We still believe that parallelisation is a good option to extract the maximum resolution of the image while reducing the detection time.

Two other alternatives can also be used to reduce the execution time while using maximum pixel resolution. The first one, that is already presented in a parallel work, is the use of the cloud capabilities. The high-resolution image is not processed in the *C&C*. Instead the App sends it to a cloud service, with high computation capabilities, to proceed with the detection with the most successful state-of-the-art models. The processed image and the total count are then returned to the agents. This solution has the advantage of broadcasting the results to more places (control centre room, other agents in the area, etc.) In contrast it has the drawback of the cyber-security threat that all this data transfer supposes. Most law-enforcement agencies are not confident with this method when the images are very sensitive.

The other alternative method is the use of drones with cameras that provide zoom capabilities. Typically zoom cameras are an expensive addition to the cost of the drone that not all local police can currently afford. Moreover, zoom can reduce crucial information about the context of the surroundings. However, we believe that, when available, zoom can help the law enforcement agent to obtain good results from our model POLO*par* by combining the adequate value of zoom with the adequate level of cropping and parallelism. For instance, if a trained model is available for images taken at 30 m, but the pilot needs to fly at 60 m, they can always zoom-in the camera as the drones were flying at 30 m and apply the validated model, being highly confident of the results.

**Author Contributions:** Conceptualization, writing and resources C.B.; supervision: E.Ç.; methodology, À.A. and J.T.; software, P.R., À.A. and J.T. All authors have read and agreed to the published version of the manuscript.

**Funding:** This research was partially funded by the AGAUR research agency of Catalonia under grant number 2020PANDE00141 and by the Ministry of Science and Education of Spain under grant number PID2020-116377RB-C21.

**Institutional Review Board Statement:** Not applicable.

**Informed Consent Statement:** Not applicable.

**Acknowledgments:** Authors would like to thank the Castelldefels Local Police, especially Modesto Sánchez, for their contribution to this work, with their images, their feedback about the prototype and the flight tests operative sharing. Additionally, our thanks to Cristina Marinescu and the *FrItPanEc* team: Matteo Conci, Jorge E. Padrón, Quentin Laurent, Freddy X. Gordillo, and Leo Conforti, Master students at the EETAC-UPC, for their help in labelling and testing many of the concepts that support this paper. Finally to our colleague Esther Salamí for the last revision.

**Conflicts of Interest:** The authors declare no conflict of interest. The funders had no role in the design of the study; in the collection, analyses, or interpretation of data; in the writing of the manuscript; or in the decision to publish the results.

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
