# Peer review of "Enhancing Drones for Law Enforcement and Capacity Monitoring at Open Large Events"

_drones, doi:10.3390/drones6110359_

Round 1
Reviewer 1 Report (Previous Reviewer 1)
No further comment
Author Response
Dear reviewer,
Thanks for the review work and for the clearance of the paper.
Best regards,
Dr. Barrado
Reviewer 2 Report (Previous Reviewer 2)
This is a revised version of a previously-rejected paper. While there is a significant amount of work described in this paper, the initial and only major objection had to do with the novelty of the authors' work.
The current manuscript is a much-improved version of the paper. The authors have made extensive corrections and additions and have gone to great lengths to argue in favor of their contribution. They also have comprehensively and politely addressed all previously-raised issues. At the very least, this demonstrates a commitment to their work.
I have some advice for the authors which hopefully will help avoid other disappointing rejections in their future work:
- the point of a scientific paper is to bring a novel contribution to the state-of-the-art (survey papers are an exception, of course). It is crucial that you argue for and demonstrate that your work is novel and that it has not been done before by others.
- additionally, you need to clearly demonstrate WHY your work is better than what has been done by others. This usually takes the form of an Evaluation section where the authors compare the results of their work to other methods from the state-of-the-art. Additionally from text descriptions, this evaluation often takes the form of tables, charts, graphs, where it is clearly visible that the presented work is better/faster/more accurate than other approaches that handle the same problem. And yes, this often involves painstakingly implementing others' work in order to do a proper and fair comparison.
Author Response
Dear reviewer,
We thank you very much for the advice and review the manuscript to better address the results and conclusions. Also, the abstract is now better focus to the contributions in efficiency.
Best regards,
Dr. Barrado
This manuscript is a resubmission of an earlier submission. The following is a list of the peer review reports and author responses from that submission.
Round 1
Reviewer 1 Report
In this article, the authors demonstrated the deep learning software developed for the local police to enhance the capabilities of drones used for the surveillance of large events. This work shows that it is possible to customize the C&C of the pilot-in-command to add new functionalities to the drone. However, there are several improvements that need to make before publication:
1) The authors should clearly list the main contributions of the work.
2) This work is developed for the surveillance of large events. In this kind of event, many people will only show parts of their body from the drone view. How did the authors solve this problem and counted the people?
3) The authors may consider adding some relative works in the introduction part for comparison, such as:
a) Luo, Cai, Leijian Yu, Jiaxing Yan, Zhongwei Li, Peng Ren, Xiao Bai, Erfu Yang, and Yonghong Liu. "Autonomous detection of damage to multiple steel surfaces from 360 panoramas using deep neural networks." Computer‐Aided Civil and Infrastructure Engineering 36, no. 12 (2021): 1585-1599.
b) Hammer, Marcus, Marcus Hebel, and Michael Arens. "Person detection and tracking with a 360 lidar system." In Electro-Optical Remote Sensing XI, vol. 10434, pp. 179-185. SPIE, 2017.
Reviewer 2 Report
The authors present what seems to be a lot of work in this paper. There is a lot of software development, training and testing of various neural networks, and detailed explanations of the resulting product.
The paper has multiple small spelling/phrasing mistakes and could use some proof-reading by an English speaker. The abstract is quite vague and does not clearly present the authors' work, only that it "enhances the capabilities of drones".
However, these issues are easily fixable. I also respect the amount of effort that the authors have put into this work and I hope that it is useful to their local police department. Disappointing as it may be, I have to say that the major problem with this manuscript is that it does not really belong in a scientific journal. The authors go into great detail when describing the software, the user interface, the predictive systems used throughout, but none of it seems to constitute original research. The authors use neural network architectures that have been developed by others. Their entire system uses already existing technologies. So the paper is not really a contribution in its respective field. Furthermore, while the authors present the results of their own method, they do not do a proper evaluation, i.e. arguing for why their method is an improvement over other approaches from the state-of-the-art. This manuscript would work well for presenting a product to potential customers (such as the police department mentioned in the paper, or developers of crowd control drones), but, despite the amount of effort described, it does not constitute a scientific paper. In future work, the authors need to answer two essential questions: 1) what makes their work original research? and 2) what makes their work an improvement over already existing results?